# Evaluation of Peri-Implant Bone Changes with Fractal Analysis

**DOI:** 10.3390/jcm14113820

**Published:** 2025-05-29

**Authors:** Nurcan Yurtoglu, Tolga Fikret Tozum, Serdar Uysal

**Affiliations:** 1Tepebasi Dental Health Center, Ankara 06290, Turkey; 1965nurcan@gmail.com; 2Department of Periodontics, Collage of Dentistry, University of Illinois at Chicago, Chicago, IL 60612, USA; ttozum@uic.edu; 3Faculty of Dentistry, Department of Oral Radiology, Hacettepe University, Ankara 06100, Turkey

**Keywords:** implant, periapical radiography, structure of bone, fractal analysis

## Abstract

**Background/Objectives:** Accurate scientific methods are essential for monitoring the osseointegration of dental implants postoperatively. This study aims to evaluate peri-implant bone changes using the fractal analysis (FA) method during follow-up. **Methods:** Periapical radiographs were obtained from 77 patients with dental implants, and 33 permanent teeth serving as a control group, retrieved from the radiology archive. Radiographs were taken using the parallel technique at 3, 6, and 12 months post-surgery. All images were digitized and saved in TIFF. Each image was aligned using the TurboReg plugin in ImageJ software. Regions of interest (ROIs) were selected from the mesial and distal aspects of the implants, then prepared for fractal analysis. FA was performed to assess changes in bone structure over time. **Results:** In the study group, radiographs of 24 patients for 0, 3 and 6 month, radiographs of 34 patients for 0, 6 and 12 month, radiographs of 8 patients for 0 and 12 month, radiographs of 5 patients for 0 and 3 month, radiographs of 5 patients for 0 and 6 month, and 1 patient of 0, 3, 6 and 12 month of radiographs were used in the study. There were no statistically significant differences in FA values over time when analyzed by gender and age in both the study and control groups. However, a statistically significant difference was observed in FA value changes over time and jaws. **Conclusions:** The study indicates a positive correlation between bone remodeling over time and FA results, likely due to the restoration of masticatory forces in the implant area. Image analysis on two-dimensional dental radiographs can be a useful tool for detecting changes in bone density. Fractal analysis is a cost-effective and practical diagnostic method for monitoring bone changes over time.

## 1. Introduction

Dental implants are artificial tooth roots, which are placed in a patient’s jawbone, produced using body compatible materials in order to eliminate the missing teeth, correct the apparent change, and restore function. The goal in implant dentistry is to regain the reduced function and esthetics without reducing the quality of life of the patient. Dental implants osseointegrate to the surrounding bone without a periodontal ligament [1]. The trabecular bone around the implant that supports the functional pressure applied by the implant plays an active role in the distribution of stress by creating a load transfer path. The key factor in a successful implant treatment is to transfer the stress to the surrounding bone [2].

In implants exposed to functional loading, bone remodeling occurs and trabecular morphology changes as bone volume, trabecular thickness and bone content increase. These changes in the bone matrix created by osteoclasts and osteoblasts alter the trabecular structure radiologically detectable. In particular, the follow-up of implant surgery and subsequent changes are quite difficult, and the most valid method is using the radiographic techniques. However, the fact that the images are two-dimensional (2D) and superpositions cause insufficiency in the measurement processes [3].

In the literature, in addition to the volume of bone structure, determining bone strength is also important [4]. Trabecular structure of bone can be analyzed by measurements such as the area of bony plates, the circumference of the trabeculae, the number of bone and bone marrow regions, the thickness of the trabeculae, the spacing of the trabeculae and the fractal size of the bone [5]. Various classification methods are available to evaluate the quality of bone. The mechanical performance of bone is the key factor for a successful implant [6]. Bone mineral density (BMD) evaluated by the dual-energy x-ray absorptiometry (DEXA), and Hounsfield Unit or bone density derived by computerized tomography have been used to determine the bone quality [7]. It is obvious that dual-energy x-ray absorptiometry is the gold standard for the evaluation of BMD; however, it is not practical for dentistry, as DEXA does not provide the cross-sectional image [8].

Fractal analysis (FA) is a statistical tissue analysis method and is the numerical expression of a mathematical pattern that shows the complexity of structures. It is based on fractal mathematics to describe complex structures and structural models and is the expression of fractal dimension (FD) by numbers [9,10].

FA is a method that expresses the roughness of the tissue by defining the self-similarity of gray-level differences in different scales of tissues and can evaluate bone density on the radiograph [8]. FDs of 2D trabecular bone projections are associated with three-dimensional trabecular bone parameters [11]. It has been shown that information about bone mineral distribution and trabecular structure can be obtained from 2D radiographs of trabecular bone [12].

FDs are correlated with biomechanical properties, and a strong correlation between demineralization of alveolar bones has been shown [13]. FD increase is localized for mechanically loaded teeth [14] and external loading stress has been shown to be the most important factor [1]. The FD around the implant after functional loading shows the remodeling of the surrounding bone and is helpful in the observation of the alveolar trabecular bone structure around the implant [1].

The aim of this study is to evaluate the peri-implant bone changes around dental implants using fractal analysis (FA) method in the follow-up.

## 2. Materials and Methods

The study protocol was approved by the Non-Interventional Clinical Research Ethics Board of Hacettepe University (Approval Number: GO 13/487-10) and was conducted in accordance with the Declaration of Helsinki and Good Clinical Practice Guidelines. Informed consent was obtained from all participants prior to radiographic evaluation. Only patients who received implant treatment at the university clinics were included in the study.

### 2.1. Obtaining Radiographs

All periapical radiographs were obtained using the same X-ray machine and standardized paralleling technique (long cone) with an aiming device, at different time intervals. Radiographs were retrospectively retrieved from the radiology archive. A total of 77 patients were included in the study based on the following criteria: absence of systemic diseases that could affect bone structure, use of the same brand of implant system, and identical implant surgical procedures. The control group consisted of 33 permanent teeth that were not adjacent to implants and had no prosthetic restorations related to implants.

Patients with at least two follow-up radiographs were included in the study, and radiographs taken using the paralleling technique during the first 10 days post-surgery (baseline), as well as at 3, 6, and 12 months, were analyzed. Radiographs were scanned using a flatbed scanner (EPSON EXPRESSION 10000 XL, Seiko Epson Co., Nagano, Japan) at 300 dpi resolution and saved in Tagged Image File Format (TIFF), an uncompressed format.

### 2.2. Image Analysis

An oral radiologist who was blinded to patient identity and follow-up period conducted the image analysis. Image enhancements such as brightness, contrast, and filtering adjustments were not permitted. All digital images were aligned using the TurboReg plugin in ImageJ (version 1.47u, NIH). The same procedure was applied to images of the control group teeth.

Considering previous studies emphasizing that the size, shape, and placement of the region of interest (ROI) can affect fractal analysis (FA) outcomes [15], these factors were standardized in this study [16]. ROIs measuring 10 × 30 pixels were placed on the mesial and distal aspects of the implants, as close as possible to the first thread of the implant body (Figure 1a). Because the images were pre-aligned, identical ROIs were used across all time periods for each patient. This approach allowed the assessment of newly formed bone and trabecular changes, which typically occur near the first implant thread.

In the control group, a reference line was drawn connecting the cemento-enamel junctions (CEJ) of the teeth. ROIs with the same dimensions (10 × 30 pixels) were placed mesially and distally from the CEJ line, consistently across all time points for each patient.

### 2.3. Fractal Analysis

During ROI selection, care was taken to exclude implant threads, adjacent teeth, lamina dura, periodontal ligament, alveolar crest, and tooth root surfaces. All selected ROIs were cropped and duplicated (Figure 1b). The duplicates were blurred using a Gaussian filter (kernel size = 35) to remove fine and medium brightness variations (Figure 1c). The blurred image was subtracted from the original (Figure 1d), and a pixel value of 128 was added to standardize brightness (Figure 1e). This image was then binarized using a threshold of 128 gray levels (Figure 1f). To reduce image noise, the binary image was first eroded (Figure 1g) and then dilated once (Figure 1h). After dilation, the perimeter of the binary image was outlined and skeletonized (Figure 1i). The resulting skeletonized image was used for FA.

In these skeletonized images, the skeletonized areas represent bone trabeculae, while non-skeletonized areas indicate bone marrow. The fractal dimension (FD) was calculated using the box-counting method. A square grid of various tile sizes (2, 3, 4, 6, 8, 12, 16, 32, and 64 pixels) was applied to each image, and the number of tiles covering trabecular bone was counted. A log–log graph was then plotted, and the slope of the line was used to calculate the FD. For each case, a single FD value was determined by averaging the FD values obtained from the mesial and distal ROIs of the implant (or tooth in the control group). FD values from baseline radiographs were compared to those from 3-, 6-, and 12-month follow-ups.

### 2.4. Statistical Analysis

Temporal changes in FA values were statistically analyzed based on time, gender, jaw (maxilla/mandible), and age. Because not all patients had radiographs at each of the four time points, mixed statistical models were used to evaluate changes over time. Differences between independent groups (e.g., by gender) were assessed using the *t*-test.

## 3. Results

The study group consisted of 48.1% females (n = 37) and 51.9% males (n = 40) between the ages of 25 and 74. Of the implants in this group, 37.7% (n = 29) are maxillary and 62.3% (n = 48) are mandibular implants. In the study group, radiographs of 24 patients for 0, 3 and 6 month, radiographs of 34 patients for 0, 6 and 12 month, radiographs of 8 patients for 0 and 12 month, radiographs of 5 patients for 0 and 3 month, radiographs of 5 patients for 0 and 6 month, and 1 patient of 0, 3, 6 and 12 month of radiographs were used in the study. Distribution of FA values according to gender, time and jaw (maxillary or mandibular) in the study and control group were presented in Table 1.

## 4. Discussion

The treatment of dental deficiencies with the help of implant-supported fixed or removable dentures is among the current treatment options; high success rates in osseointegrated implants have made these implants widely used and become the primary treatment option [17]. The key factor in a successful implant treatment is the ability to transfer stress from the chewing forces to the surrounding bone. The link between modeling and remodeling is critical in maintaining stable implant-bone surfaces after loading [2]. If the loading forces can be controlled properly, new bone formation around the implant is stimulated, which helps to keep the implant stable [18].

Imaging of the bone formation around the implant is important in the follow-up and various clinical and radiological imaging techniques are used to monitor the success of osseointegration. Periapical radiographs, which were obtained with paralleling technique, are frequently used in the follow-up of bone loss in the mesial and distal surfaces of the implant. However, it only shows marginal bone loss on the mesial and distal surfaces [19]. Unfortunately, this frequently used method has low sensitivity and has a limited diagnostic value to detect early changes in bone [20].

The cage-like structure observed on intraoral radiographs is the image of the medullar cavity of the trabecular bone. Various measurement methods have been developed for its identification. These methods identify the area of the bony plates, the number of bone and bone marrow regions, the thickness and spacing of the trabeculae, the circumference of the trabeculae, and the FD of the bone [5]. The FD of the bone is a mathematical method that is considered more valid than other methods [12,21,22]. In the present study, FD was used for measurement purposes in order to obtain numerical data.

FA is one of the analytical techniques used to evaluate bone quality and it is associated with changes in bone density and reflects partial demineralization of bone [23]. Numerous studies have demonstrated its usefulness in analyzing biological images. FA offers a quantitative method to detect changes in bone structure. It is unaffected by variables such as projection geometry or radiation dose, and it is a non-invasive, accessible, and cost-effective method. As a result, its use in medicine and dentistry continues to grow [24]. Its validity and reliability have been reported in many dental studies, especially in evaluating bone structure [25]. The image of the internal structure of the alveolar bone resembles a lattice formed by thin spicules, trabeculae, and lamellae. Trabecular bone features a branching structure that exhibits fractal characteristics, such as self-similarity and scale invariance. Therefore, measuring FD using fractal geometry can help assess trabecular complexity and bone structure. This allows information on subtle, otherwise invisible, changes in the three-dimensional trabecular bone to be obtained from two-dimensional images using FA values.

In dentistry, FA has been shown to quantitatively detect changes in the alveolar bone, in the trabecular architecture of the mandibular condyle in patients with temporomandibular joint disorders. The use of FA in dentistry has increased substantially in recent years. Quantification of trabecular bone is required in several dental specialties, and the use of common dental images such as panoramic and periapical radiographs is referred [26]. The jaw-related FD sensitivity in osteoporosis screening was found to be a better parameter [27]. Significant changes in trabeculation of the condyle and mandibular corpus were found during functional orthodontic treatment on dental panoramic radiographs [28]. Bulut and Tokuc [29] showed that fractal analysis from panoramic radiographs provides additional information about the status of the trabecular structure of the mandibular condyles of children and the FD values of the mandibular condyle trabecular structure varies with age.

Soltani et al. [30] mentioned that FA is a useful tool for evaluation of bone alterations in moderate and severe periodontitis. Yarkac et al. [31] showed that the FD values of trabecular bone are different in healthy individuals and individuals with different stages of periodontitis. FA can quantitatively and objectively detect alterations in the interdental trabecular pattern of patients with periodontitis.

Celebi et al. [32] demonstrated changes in the jawbones of patients with RA using image analysis methods to quantify trabecular parameters. This assessment provided an objective evaluation of the bone. Analyses of panoramic radiographs, taken during routine dental examinations, can reveal bone changes in patients with RA and thereby offer insights into disease prognosis. Early detection of these impacts on the bone can improve the quality of life for these patients.

Jolley et al. [33] showed that periapical radiographs are a reliable method to detect changes in various bone diseases by FD analysis. Bollen et al. [34] performed FA to evaluate the bone quality and obtained significant diagnostic results. FA is a statistical tissue analysis method based on fractal geometry to identify complex structures and structural models. It is expressed as numbers in FD [9,34]. FA defines the trabecular bone on a different scale in a self-similar detail. Trabecular bone has a branching structure that exhibits a unique fractal feature. In this structure, FDs can be measured and fractal geometry can be used to determine trabecular complexity and bone structure [35]. There are several methods to calculate the FD, and box counting method, which is defined by Russell et al. [36], is the most frequently used method in the medical field. With the box counting algorithm, trabecular structure is measured by counting the trabecular bone and bone marrow interface and evaluate the boundaries of the trabecular bone and bone marrow [37]. The high values determined as a result of box counting represent the numerical algorithm of a more complex structure [38,39]. In accordance with these data obtained from the literature, box-counting method was used in the present study.

Chen and Chen [40] emphasized that FD is low in (less complex) large trabecular diameter structures. FD values are larger in the maxilla than in the mandible due to its structural features in periapical radiographs [41]. Interestingly, the present study results demonstrated that the FA values in the mandible were higher than the values in the maxilla. Depending on the inequality of the measurements in the maxilla and mandible, the minimum values of the measurements in jaws were found close to each other. However, the maximum values were found higher in the mandible. Because of these results, we assume that our FA values in the mandible are higher.

The studies showed that as bone density increased, the complexity of spongious structures and FA values increased. In addition, FD values decreased as the bone density FA values decreased [42]. It was also reported that FD increased during the healing process of the bone especially in the neck area of the implant. FD values decreased after application of the implant and increased gradually according to the bone healing process [38]. Similar results were found in the present study, and the increase in FD values over time was found to be consistent with the literature.

FA is not sensitive to small differences in x-ray parameters, angulation of x-ray and ROI position, and is a reliable method in the analysis of trabecular bone density changes in non-standard periapical radiographs [33,43]. On the contrary, FD could be affected by noise generated from the imaging process and is affected by the shape and size of the ROI [15]. Therefore, in the present study, radiographs were aligned before ROIs were obtained. Thus, the same ROIs were always set. The diversity of results defined in the literature is due to differences in measuring FD and selecting the area [44].

It was emphasized that FD is a useful morphometric identifier showing the quality of bone [45,46]. However, some researchers point out that the results of the different methods used to calculate FD may not match. Many researchers have stated that this technique may be the descriptor of bone structure but their studies cannot be compared because they do not use the same algorithm or ROI [43,45,46]. Researchers acknowledge that there is a mismatch in fractal values calculated and compared between studies [47].

However, despite all these different ideas, the FA of bony tissue is defined as an accurate, economical, and easily applicable method [38]. It is emphasized that FA is a non-invasive tool that can be used in clinical studies and is not affected by irradiation and small positioning errors [48]. As suggested previously, FA may be a useful method for understanding the healing process around implants [18].

## 5. Conclusions

The results of the present study imply that after the implant application, there is a positive relationship between the changes in the bone over time and the FA results due to the regained chewing forces in the bone area. Image analysis procedures performed on 2D dental radiographs may be used to detect changes in bone density. Being able to make three-dimensional evaluations using 2D images provides an advantage in terms of patient safety since less radiation is used. FA is an inexpensive and practical diagnostic tool that can be used to track changes in the bone. These changes in the bone become more measurable with FA applied on radiographs taken during the control phases. With the help of a macro written for image j, fractal analysis values can be obtained from the periapical radiographs taken at the control appointments of the patients, and the change in bone complexity between the control periods can be monitored.

## Figures and Tables

**Figure 1 jcm-14-03820-f001:**
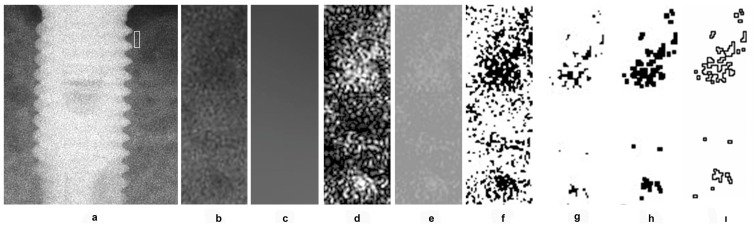
Application of fractal analysis. (**a**) Selection of ROI; (**b**) ROI; (**c**) remove fine and medium brightness variations; (**d**); blurred image subtracted from the original; (**e**) pixel value of 128 added to standardize brightness; (**f**) binarized using a threshold of 128 gray levels; (**g**) the binary image eroded; (**h**) the binary image dilated; (**i**) binary image outlined and skeletonized.

**Table 1 jcm-14-03820-t001:** Distribution of FA values according to gender, time, and jaw in the study and control group.

	Study Group						Control Group				
		N	Mean	SD	Min	Max	N	Mean	SD	Min	Max
**Sex**	**Women**	102	1.6073	0.2102996	1.3533	1.8359	50	1.6659	0.2143166	1.3537	1.8320
	**Men**	113	1.5813	0.1845743	1.3595	1.8352	46	1.6450	0.2166000	1.3596	1.8318
	**Total**	215	1.5937	0.1971607	1.3533	1.8359	96	1.6559	0.2145322	1.3537	1.8320
**Time**	**0. Month**	77	1.5883	0.1944352	1.3533	1.8319	33	1.6537	0.2162132	1.3549	1.8319
	**3. Month**	31	1.4942	0.1591801	1.3596	1.8314	9	1.4509	0.1772831	1.3596	1.8314
	**6. Month**	64	1.5958	0.2047934	1.3596	1.8337	31	1.6516	0.2219899	1.3552	1.8320
	**12. Month**	43	1.6718	0.1871479	1.3625	1.8359	23	1.7450	0.1630542	1.3537	1.8319
	**Total**	215	1.5937	0.1971607	1.3533	1.8359	96	1.6559	0.2145322	1.3537	1.8320
**Jaw**	**Maxilla**	82	1.4789	0.1623702	1.3549	1.8319	31	1.4603	0.1705796	1.3549	1.8316
	**Mandible**	133	1.6644	0.1833463	1.3533	1.8359	65	1.7492	0.1652641	1.3537	1.8320
	**Total**	215	1.5937	0.1971607	1.3533	1.8359	96	1.6559	0.2145322	1.3537	1.8320

In the study group, between 0- and 3-months’ FA values (Wald X^2^ = 4.386, *p* = 0.036) and between 0 months and 12 months’ FA values (Wald X^2^ = 4.600, *p* = 0.032) were statistically significant. No statistically significant difference was found between the baseline and 6 months FA values (*p* > 0.05). There was no statistically significant difference in the study and control groups when the change in FA values by time was evaluated by gender and age (*p* > 0.05). In the study and control group, a statistically significant difference was detected in the change by time and jaws (Wald X^2^ = 19.151, *p* = 0.0001; Wald X^2^ = 23.431, *p* = 0.0001, respectively).

## Data Availability

A consent form was obtained from the participants while they were included in the study. In this consent form, a form was signed stating that the data would not be shared for confidentiality. Since the ethics committee’s approval was obtained in this way, it is not possible to share the data without consulting the ethics committee.

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
