# Peer review of "Evaluation of Peri-Implant Bone Changes with Fractal Analysis"

_jcm, 2025, doi:10.3390/jcm14113820_

Round 1
Reviewer 1 Report
Comments and Suggestions for Authors
Dear Authors,
This paper addresses an interesting topic. I would recommend several modifications. Below these are some suggestions for You:
- Abstract: Should be written again. ‘Sth is important’ is not what should be written. The abstract should be structured, but without subheadings. Another important matter is English.
- There are significant flaws throughout the methods. This paragraph should be structured (with subheadings). Eligibility criteria are needed.
- Lines 131-132: Skeletonized? Skeletonized binary image? This term should be explained
- Table 1 is doubled. Delete the second one.
Best regards and good luck
Author Response
REVIWER 1
This paper addresses an interesting topic. I would recommend several modifications. Below these are some suggestions for You:
- Abstract: Should be written again. ‘Sth is important’ is not what should be written. The abstract should be structured, but without subheadings. Another important matter is English.
The abstract was rewritten as indicated by the reviewer and the English version was edited by a native English speaker. Also highlighted yellow in the revised manuscript.
- There are significant flaws throughout the methods. This paragraph should be structured (with subheadings). Eligibility criteria are needed.
Materials and Methods was structured with subheadings and highlighted yellow in the revised manuscript.
- Lines 131-132: Skeletonized? Skeletonized binary image? This term should be explained.
“Skeletonized binary image” term was chanced as “Skeletonized image” in lines 131-132. Skeletonization is a stage used in fractal analysis. where skeletonized areas represent bone trabeculae, while non-skeletonized areas indicate bone marrow.
- Table 1 is doubled. Delete the second one.
Second Table 1 was deleted.
Reviewer 2 Report
Comments and Suggestions for Authors
ABSTRACT
1.“In results, it’s better to describe the results of different time points as mentioned in methods that 3, 6, and 12 months”
INTRODUCTION
- Based on what's mentioned in the key words and abstract, FA is the key point in this study, but it is a little bit confused that most parts of this manuscript mention FD, please make it more clearly.
MATERIALS AND METHODS
- Please make the description of the ROI selection could be more precise and concise. The paper states, "These features were considered" and "In order to determine the FA value of the newly formed bone and the change over time," separately explaining the reason for selecting the ROI at "the first thread of the implant."
- Please unify the image number and the content in the manuscript.”
- Please explaine the reason to choose the pixel size in box counting as 2, 3, 4, 6,8, 12,16, 32 and 64, and which pixel size was used to calculate the FD value.”
- The two tables in the manuscript is same, and is it a typo?
DISCUSSION
- Please explain the reason why the result is different from others’ in paragraph Line 209.
REFERENCE
9. The nearest reference is in 2013, please add more in recent 5 years.
Author Response
REVIEWER 2
Comments and Suggestions for Authors
ABSTRACT
1.“In results, it’s better to describe the results of different time points as mentioned in methods that 3, 6, and 12 months”
In the results section of the Abstract, results from different time periods are added and highlighted in yellow.
INTRODUCTION
- Based on what's mentioned in the key words and abstract, FA is the key point in this study, but it is a little bit confused that most parts of this manuscript mention FD, please make it more clearly.
Fractal analysis (FA), is a statistical tissue analysis method and is the numerical expression of a mathematical pattern that shows the complexity of structures. It is based on fractal mathematics to describe complex structures and structural models, and is the expression of fractal dimension (FD) by numbers.
This sentence reorganized in the Introduction and highlighted yellow. We hope we could able to clear up the confusion.
MATERIALS AND METHODS
- Please make the description of the ROI selection could be more precise and concise. The paper states, "These features were considered" and "In order to determine the FA value of the newly formed bone and the change over time," separately explaining the reason for selecting the ROI at "the first thread of the implant."
In the Materials and Methods, how ROI selection was done is described in more detail and highlighted in yellow.
- Please unify the image number and the content in the manuscript.”
The figure number and application of fractal analysis were made visible in the figure.
- Please explaine the reason to choose the pixel size in box counting as 2, 3, 4, 6, 8, 12, 16, 32 and 64, and which pixel size was used to calculate the FD value.”
The box counting method is a method used to calculate the fractal dimension. In this method, the number of pieces is calculated as the number of boxes sufficient to completely scan a shape. In practice, it is determined by counting the area of regular boxes placed on the object covering the object.
The fractal dimension is calculated based on the gradient (variation) of the box sizes. A square grid of various tile sizes (2, 3, 4, 6, 8, 12, 16, 32, and 64 pixels) was applied to each image (all tile sizes used), and the number of tiles covering trabecular bone was counted. A log–log graph was then plotted, and the slope of the line was used to calculate the FD. For each case, a single FD value was determined by averaging the FD values obtained.
- The two tables in the manuscript is same, and is it a typo?
The extra uploaded Table 1 was deleted.
DISCUSSION
- Please explain the reason why the result is different from others’ in paragraph Line 209.
There are various techniques to perform fractal analysis. In the medical field, the box counting method developed by Russel et al. is preferred due to its ease of application. Therefore, we preferred this method in our study. Different fractal dimensions are obtained depending on the method used. Therefore, different results are encountered.
REFERENCE
- The nearest reference is in 2013, please add more in recent 5 years.
Since the fractal analysis technique is a previously developed technique, we have used older sources to cite the main references, but current references have been added to our References as requested by the reviewer. We thank the Reviewer for his attention and rightful guidance.
Round 2
Reviewer 1 Report
Comments and Suggestions for Authors
Dear Authors,
I have no further requests.
Regards